# Development and External Validation of Deep-Learning-Based Tumor Grading Models in Soft-Tissue Sarcoma Patients Using MR Imaging

**DOI:** 10.3390/cancers13122866

**Published:** 2021-06-08

**Authors:** Fernando Navarro, Hendrik Dapper, Rebecca Asadpour, Carolin Knebel, Matthew B. Spraker, Vincent Schwarze, Stephanie K. Schaub, Nina A. Mayr, Katja Specht, Henry C. Woodruff, Philippe Lambin, Alexandra S. Gersing, Matthew J. Nyflot, Bjoern H. Menze, Stephanie E. Combs, Jan C. Peeken

**Affiliations:** 1Department of Radiation Oncology, Klinikum Rechts der Isar, Technical University of Munich (TUM), Ismaninger Straße 22, 81675 Munich, Germany; fernando.navarro@tum.de (F.N.); hendrik.dapper@mri.tum.de (H.D.); rebecca.asadpour@tum.de (R.A.); stephanie.combs@tum.de (S.E.C.); 2Department of Informatics, Technical University of Munich (TUM), Boltzmannstr. 3, 85748 Garching, Germany; bjoern.menze@tum.de; 3TranslaTUM—Central Institute for Translational Cancer Research, Einsteinstraße 25, 81675 Munich, Germany; 4Department of Orthopedics and Sports Orthopedics, Klinikum Rechts der Isar, Technical University of Munich (TUM), Ismaninger Straße 22, 81675 Munich, Germany; carolin.knebel@mri.tum.de; 5Department of Radiation Oncology, Washington University in St. Louis, 4511 Forest Park Ave, St. Louis, MO 63108, USA; mspraker@wustl.edu; 6Department of Radiology, Grosshadern Campus, Ludwig-Maximilians-University Munich, Marchioninistraße 15, 81377 Munich, Germany; vincent.schwarze@med.uni-muenchen.de (V.S.); alexandra.gersing@tum.de (A.S.G.); 7Department of Radiation Oncology, University of Washington, 1959 NE Pacific St, 356043, Seattle, WA 98195, USA; skschaub@uw.edu (S.K.S.); ninamayr@uw.edu (N.A.M.); nyflot@uw.edu (M.J.N.); 8Department of Pathology, Technical University of Munich (TUM), Trogerstr. 18, 81675 Munich, Germany; katja.specht@tum.de; 9Department of Precision Medicine, GROW—School for Oncology and Developmental Biology, Maastricht University, Universiteitssingel 40, 6229 ER Maastricht, The Netherlands; h.woodruff@maastrichtuniversity.nl (H.C.W.); philippe.lambin@maastrichtuniversity.nl (P.L.); 10Department of Radiology and Nuclear Imaging, GROW—School for Oncology and Developmental Biology, P. Debyelaan 25, 6229 HX Maastricht, The Netherlands; 11Department of Radiology, Klinikum rechts der Isar, Technical University of Munich (TUM), Ismaninger Straße 22, 81675 Munich, Germany; 12Department of Radiology, University of Washington, 4245 Roosevelt Way NE, Seattle, WA 98105, USA; 13Department for Quantitative Biomedicine, University of Zurich, Winterthurerstrasse 190, CH-8057 Zurich, Switzerland; 14Institute of Radiation Medicine (IRM), Department of Radiation Sciences (DRS), Ingolstaedter Landstr. 1, 85764 Munich, Germany; 15Deutsches Konsortium für Translationale Krebsforschung (DKTK), Partner Site, 85764 Munich, Germany

**Keywords:** deep learning, convolutional neural networks, artificial intelligence, machine learning, soft-tissue sarcomas, tumor grading, MRI

## Abstract

**Simple Summary:**

In soft-tissue sarcoma (STS) patients, the decision for the optimal treatment modality largely depends on STS size, location, and a pathological measure that assesses tumor aggressiveness called “tumor grading”. To determine tumor grading, invasive biopsies are needed before therapy. In previous research studies, quantitative imaging features (“radiomics”) have been associated with tumor grading. In this work, we assessed the possibility of predicting tumor grading using an artificial intelligence technique called “deep learning” or “convolutional neural networks”. By analyzing either T1-weighted or T2-weighted MRI sequences, non-invasive tumor grading prediction was possible in an independent test patient cohort. The results were comparable to previous research work obtained with radiomics; however, the reproducibility of the contrast-enhanced T1-weighted sequence was improved. The T2-based model was also able to significantly identify patients with a high risk for death after therapy.

**Abstract:**

Background: In patients with soft-tissue sarcomas, tumor grading constitutes a decisive factor to determine the best treatment decision. Tumor grading is obtained by pathological work-up after focal biopsies. Deep learning (DL)-based imaging analysis may pose an alternative way to characterize STS tissue. In this work, we sought to non-invasively differentiate tumor grading into low-grade (G1) and high-grade (G2/G3) STS using DL techniques based on MR-imaging. Methods: Contrast-enhanced T1-weighted fat-saturated (T1FSGd) MRI sequences and fat-saturated T2-weighted (T2FS) sequences were collected from two independent retrospective cohorts (training: 148 patients, testing: 158 patients). Tumor grading was determined following the French Federation of Cancer Centers Sarcoma Group in pre-therapeutic biopsies. DL models were developed using transfer learning based on the DenseNet 161 architecture. Results: The T1FSGd and T2FS-based DL models achieved area under the receiver operator characteristic curve (AUC) values of 0.75 and 0.76 on the test cohort, respectively. T1FSGd achieved the best F1-score of all models (0.90). The T2FS-based DL model was able to significantly risk-stratify for overall survival. Attention maps revealed relevant features within the tumor volume and in border regions. Conclusions: MRI-based DL models are capable of predicting tumor grading with good reproducibility in external validation.

## 1. Introduction

Soft-tissue sarcomas (STS) constitute a rare cancer type [1]. Risk stratification is primarily performed using tumor location, pathological tumor grading, tumor size, and certain histological subtypes [2]. One of the most decisive factors constitutes tumor grading. Two separate grading systems were originally defined by the French Federation of Cancer Centers Sarcoma Group (FNCLCC) and the National Cancer Institute (NCI) [3,4]. The FNCLCC system, however, showed better predictive values for distant metastases and is used predominantly worldwide [5]. While FNCLCC G1 (termed “low-grade”) STS are generally treated with surgery alone, FNCLCC G2/G3 (termed “high-grade”) STS require multi-modal therapy regimens involving radiotherapy and/or chemotherapy [6,7,8]. Despite treatment intensifications, the overall outcome remains poor for high-grade STS [9,10,11]. 

Quantitative imaging constitutes an alternative method to characterize tissues. In contrast to a focal biopsy sample, image analysis is capable of assessing the whole tumor volume and can enable longitudinal assessment. In recent years, two general analysis methods have been developed which are summarized under the term “radiomics” [12]. First, predefined handcrafted features are extracted by analyzing the tumor’s shape, intensity distribution, and texture. Afterwards, machine learning models are applied to predict clinical endpoints [13,14,15,16]. Second, approaches such as neural networks can be specifically trained to directly analyze imaging data to make end-to-end predictions [17]. Convolutional neural networks (CNNs) describe a class of architectures that are especially suited for image analysis and that are, among others, often referred to by the terms “deep learning” (DL) or “artificial intelligence” (AI). In DL, the systems can be categorized into supervised, unsupervised, and semi-supervised learning according to their learning strategy. In this work, we used deep supervised learning, where the neural network requires annotations to learn discriminative features directly from the images without need of extra information. Since the input features are the raw images, there is no need for feature extraction or feature selection as in traditional machine learning. Both techniques (traditional machine learning and DL) have been shown to predict prognosis, tumor progression, molecular aberrations, or spatial infiltration in various cancer subtypes [18,19,20,21,22,23,24]. Some studies found superior predictive performances using CNNs compared to handcrafted features [25,26]. Radiomics-based approaches also enable localization and segmentation of volumes of interest (VOI) [27,28]. 

In STS patients, multiple groups previously demonstrated the possibilities of radiomics and DL to predict patients’ prognosis based on MRI, CT, and PET imaging [29,30,31,32,33,34,35]. Wang et al. developed radiomic models to differentiate malignant and benign soft-tissue lesions [36]. Further research studies evaluated the differentiation of high-grade from low-grade STS based on MRI and CT imaging scans using radiomic analysis [37,38,39,40,41,42]. No study has yet analyzed the possibility of DL-based tumor grading prediction. 

The scope of this study was to evaluate the potential of DL to predict tumor grading based on pre-therapeutic MRI scans. The value of T2-weighted fat-saturated (T2FS) MRI sequences was compared to contrast-enhanced and fat-saturated T1-weighted (T1FSGd) MRI sequences. All models were externally validated and tested for significant patient risk stratification. Attention maps were generated to evaluate relevant qualitative imaging features and increase explainability of the developed models.

## 2. Materials and Methods

### 2.1. Patients

Two independent consecutive patient cohorts from the Technical University of Munich, Munich, Germany (TUM) and the University of Washington, Seattle, WA, USA (UW) were collected retrospectively. Inclusion criteria included: histologically proven STS with available FNCLCC tumor grading information. Exclusion criteria were endoprosthesis-dependent MRI artifacts, previous radiotherapy, primary bone sarcomas, or Ewing sarcomas. Patient records were analyzed for FNCLCC tumor grading and basic patient demographics. The patient cohort with a higher balance between low-grade and high-grade STS was selected for training (TUM). In the training cohort, for each sequence all available patients were included (T1FSGd: 148 patients, T2FS: 130 patients). To allow a better comparison in the test set (UW), all patients that did not have both MRI sequences were excluded (final test set: 158 patients). 

See Appendix A for a patient workflow. In the final patient cohort, no modeling-specific data were missing. Overall survival (OS) was calculated from the initial pathologic diagnosis to the time point of death or the time point of censoring. Data reporting follows the STARD recommendations (Appendix A: STARD checklist) [40]. 

### 2.2. Image Acquisition, Definition of Volumes of Interest and Preprocessing

Pre-therapeutic MRI scans were analyzed for each included patient. See Appendix A for acquisition parameters and scan planes. For all STS, tumor segmentation was performed using Eclipse 13.0 (Varian Medical Systems, 3100 Hansen Way, Palo Alto, CA 94304, USA), MIM software version 6.6 (MIM Software Inc., 25800 Science Park Dr #180, Beachwood, OH 44122, USA), iplan RT 4.1.2 (Brainlab, Olof-Palme-Straße 9, 81829 Munich, Germany), and 3D Slicer (3D Slicer, version 4.8 stable release). The primary tumor as the VOI was manually segmented by JCP, by adapting existing expert segmentations from RT treatment planning in the TUM cohort. In the UW cohort, segmentation was performed by MBS, MM, JCP, and TC. Edematous changes were not included in the VOI. N4ITK MRI bias field correction was applied to each imaging study using the Slicer3D implementation to compensate for non-uniform intensity caused by field inhomogeneity [43]. 

### 2.3. Data Preprocessing

All volumes were resampled to $1\;{\mathrm{mm}}^{3}$ isotropic resolution and normalized using z-score normalization. From the 3D VOI, transversal 2D slices were obtained and resized to 224 × 244 before sending the images to the deep neural networks, according to the requirements of the pre-trained architecture for the 2D model. Obtaining transversal slices from one patient allowed us to increase the number of training samples for the deep neural networks. This means that from every patient in the training set we can generate as many training samples as transversal slices are available from the patient tumor. When counting the overall number of training samples, we can then go from hundreds in the original MRI data to thousands after slicing the patient. 

### 2.4. MRI-Based DL Models

We developed DL models to differentiate low-grade (G1) and high-grade (G2/3) STS. For each sequence, a separate DL model was developed: *DL-T1FSGd* and *DL-T2FS*. The base deep learning architecture for this study was based on the ImageNet pre-trained DenseNet-161 described in Figure 1 [44].

We empirically found that other architectures, including VGGNet [44], ResNet [26], WideResNet [45], AlexNet [46], and CBRNet [47] in 2D and 3D resulted in worse-performing models for our task. DenseNet 161 was the optimal architecture for tumor grading during optimization. Other architectures as well as deeper or shallower pre-trained networks obtained sub-optimal results. Similarly, other approaches such as full MR image without VOI selection, VOI image masked with tumor segmentation, and VOI image and mask as extra channel were tested with inferior performance compared to the proposed approach.

### 2.5. Optimization of Deep Learning Models

All models were developed in Pytorch with a 12 GB Titan XP [48]. The models were trained with a batch size of 30 and a learning rate of $1\times 10^{-4}$ with an ADAM optimizer for 100 epochs. We used early stopping during training, monitoring the validation loss to select the best model. Categorical weighted cross-entropy was used as the loss function. Data augmentation was applied at training time and included vertical and horizontal flip, random rotation, random zoom, elastic transform, and random cropping. Additional training details and the code can be found online (https://github.com/ferchonavarro/SarcomaTumorGrading) (accessed on 4 June 2021).

### 2.6. Evaluation Strategy

To evaluate the performance, reproducibility, and generalizability of the MRI-based DL models, stratified 5-fold cross-validation with 3 repetitions was performed, producing 15 DL models per image modality. For training and validation of the DL models, the TUM patient cohort was used (referred to as “training cohort”). All 15 models were externally tested using the UW cohort (referred to as “testing cohort”). During inference time, to obtain the tumor grading prediction per patient, the average of all 15 models and all transversal slices in the VOI was computed. Finally, the soft-max activation function converted the average predicted values into the probabilities of low-grade and high-grade STS.

### 2.7. Interpretability of DL Models

Visualization of attention maps is shown together with the model probabilities to further gain insights on the model predictions for tumor grading. The attention maps were obtained from gradient-weighted class activation maps (Grad-CAM) [49]. 

### 2.8. Comparison to Baseline Models

To compare the clinical relevance of the developed models, we compared the DL-based models to regression models using clinical features (TNM T-stage, TNM-N-stage, TNM M-stage, Age) (*Clinical*), tumor volume (*Tumor-Volume*), and the combination of clinical features and tumor volume (*Clinical-Volume-Combined*). The same aforementioned strategy was used for model evaluation.

### 2.9. Statistical Analysis

Statistical analysis and modeling were performed using Python 3.6. Model performances were characterized using calibration curves, receiver operating characteristic curves (ROC), and additional classification metrics. In addition, 95% confidence intervals were generated using 1000-fold bootstrapping. Kaplan–Meier survival curves were used to analyze model-based stratification for OS in the test set. The maximum argument from the probabilities was used to split patients into low-risk and high-risk patients. Statistical significance was tested using the log-rank test. Bonferroni correction was performed in cases of multiple testing as specified. A p-value below 0.05 was regarded as significant.

## 3. Results

### 3.1. Patient Characteristics, Histology, and VOI Definition

Overall, patient demographics were similar (Table 1). However, the distribution of histology subtypes and patients’ age was significantly different between both cohorts (*p* < 0.001, *p* = 0.03) (Appendix A). Moreover, the training cohort consisted of 35.1% low-grade and 64.9% high-grade STS. The testing cohort showed a more uneven distribution with 14.5% low-grade and 85.5% high-grade STS. 

### 3.2. Classification Performance

The results shown in Figure 2 describe the ROC curves and AUCs for the baseline models and DL-based models classifying patients as low or high-grade STS in the independent test set. It can be observed that for the baseline models (Clinical, *Tumor-Volume*, *Clinical-Volume-Combined*) the obtained AUCs were 0.54, 0.59, and 0.57, respectively. In contrast, the developed DL-based models achieved AUC values of 0.75 and 0.76 for DL-T1FSGd and DL-T2FS, respectively. Table 2 depicts additional classification metrics. All models showed good precision of at least 0.87. DL-T1FSGd classified with the best accuracy of 0.83. This was also reflected by the best sensitivity value of 0.91 but with a suboptimal specificity of 0.40. Delta-T2FS had a better specificity of 0.72 but with the cost of a worse sensitivity value of 0.62, leading to a total accuracy of 0.64. In terms of the less imbalance-biased metric, F1-Score Delta-T1FSGd achieved the best result (0.90). See Appendix A for calibration curves.

### 3.3. Patient Risk Stratification

We used the classification of the developed DL-based grading models for dichotomization of the patient cohort into low-risk and high-risk patients to evaluate the stratification performance for OS. In Figure 3, the Kaplan Meier (KM) survival curves and results of the log-rank test for the baseline models (*Clinical*, *Tumor-Volume*, *Clinical-Volume-Combined*), the ground truth tumor grading stratification (*Grading*), and the DL-based models are shown. *Clinical* and *Grading* achieved significant patient stratification (*p*-value = 0.028 and 0.02, respectively). We also found that both DL-based models separated survival curves into low-risk and high-risk patients. However, only the *DL-T2FS* achieved significant patient stratification (*p*-value = 0.045).

### 3.4. Prediction Visualization and Model Interpretability

Figure 4 and Figure 5 depict representative attention maps for the *DL-T1FSGd* and *DL-T2FS* models. Four general patterns can be observed: (1) in many cases, the largest area of activation was present within the tumor volume depicting the tumors’ “texture” (e.g., Figure 5a,b); (2) the second most frequent activation was seen in border areas of the tumor focusing on good or bad confinement (e.g., Figure 4a); (3) within border areas the interfaces of tumor to bone and tumor to vessel were frequently represented (Figure 4b,c); (4) in a small number of false cases the network failed to locate the tumor on the cropped image, focusing instead on normal anatomy or air (Figure 5d). Patterns 1–3 were often seen in parallel on the same slice (e.g., Figure 4d) or on different slices of the same tumor. 

## 4. Discussion

In this work, we developed DL-based tumor grading models based on two distinct MRI sequences. The T2FS-based DL model achieved the best predictive performance in an independent testing cohort, comparable to a previously published radiomic model. The contrast-enhanced T1-based model achieved a better performance than a previously published model. The T2-based model was able to significantly risk-stratify STS patients for overall survival. Attention maps confirmed tumor-specific features within and surrounding the tumor volume.

In a previous study, we used similar patient cohorts to develop and externally test radiomics-based tumor grading models [37]. For T2FS with AUC values of 0.76 (DL) and 0.78 (handcrafted features), the predictive performance was comparable, although with a slightly higher performance of the handcrafted feature model in the test set. For T1FSGd, the DL model showed a higher performance with an AUC of 0.75 while the handcrafted feature model achieved an AUC of 0.69. It should be noted that both cohorts have been expanded since then. The skewed proportion of low-grade and high-grade STS, however, remained similar. The patient numbers in the training set size were enlarged by 6% and 20%, and in the test set by 53% and 53% for T2FS and T1FSGd, respectively. The training set size also played a role in the comparison of our DL models. To allow direct comparability, we selected only patients for the test set that had both imaging studies available. *DL-T2FS* had a 12% smaller training sample number than the *DL-T1FSGd* model. *DL-T2FS* achieved a higher AUC but worse classification performance (e.g., F1-Score) than *DL-T1FSGd*. A previous study, however, showed a correlation between DL model performance and training sample size on a logarithmic scale [51]. Thus, for significant model performance improvements, much larger differences in training size would be beneficial and a large impact of the small differences in training size is rather unlikely.

As previously mentioned, other authors have evaluated tumor grading prediction using MRI-based radiomics [38,39,40,41,42]. However, only one study validated their models in an external testing cohort [40]. In this study, Yan et al. used a training cohort of 109 patients to develop radiomic models based on T2FS and T1-weighted MRI sequences (without contrast-enhancement). In the 70-patient test set, both models achieved predictive performances with AUCs of 0.645 (T2FS) and 0.641 (T1). Combining both features significantly increased the performance up to an AUC of 0.829. In contrast, our study used contrast-enhanced fat-saturated T1-weighted MRI scans. Both developed models had better performances than the single sequence models but were inferior to the combined model, although in a similar range. Interestingly, an additive benefit following a combination of the radiomic feature sets of both sequences (T1FSGd and T2FS) could not be observed in our previous study. However, the testing cohort was significantly smaller, increasing the chance-based risk of falsely optimistic or pessimistic results. Moreover, it had a more balanced distribution of low-grade and high-grade STS. This may also explain the lack of significant patient risk stratification of the combined model by Yan et al. Still, combining multiple imaging modalities for DL models remains a promising approach. 

The attention map analysis gave insights into the functioning of the DL models. This allows a certain amount of semantic explainability which cannot be derived for models based on handcrafted features. In many cases, the DL model focused on the internal texture structures of the STS. This may correspond to features implemented in the previously published radiomic models that were always restricted to the gross tumor volume as VOI. At the same time, it may represent semantic imaging features, such as necrosis, that have previously been linked with tumor grading [52]. Interestingly, our DL models also regularly focused on tumor-surrounding tissue, reflecting, e.g., the confinement of the tumor-tissue border. In accordance, another semantic feature, “peritumoral enhancement”, has previously been described as being correlated with tumor grading [52]. Further work is needed to evaluate associations between attention maps and semantic features. 

In a select number of cases the model did not correctly locate the tumor but instead focused on unrelated areas (e.g., air), restricting a reliable prediction. These cases predominantly occurred in rare anatomic locations (e.g., location at the trunk in Figure 5d) and constitute a limitation when extending the cropped images beyond the VOI. By increasing future training sample sizes, or excluding rare anatomical sites, the resulting models might learn to perform better. By providing the attention maps alongside each prediction, the physician could directly assess the technical reliability of the prediction. A future direction to use attention maps could be to objectively identify regions of concern for a high risk of positive margins as well as potential internal sub-volumes of high-grade histology that might inform design of future risk-adaptive, precision clinical trials for spatial intensification of therapies.

As in many STS studies, a large plethora of histologies was combined to achieve significantly large patient cohorts. As these different subtypes stem from different mesenchymal tissue types, one could speculate that histology-specific models may be more effective in predicting histology-specific tumor grading. Sub-cohorts of patients with relevant histological groups such as pleomorphic sarcomas or with dominant myxoid or fibrous matrix comprise only 19–56 patients in the training set. Previous research demonstrated a significant decrease in classification performance below 100 samples [53]. This would further be aggravated by the cross-validation approach, a low event-rate, and missing imaging scans. As a consequence, no histology-specific models could be effectively trained using the underlying patient cohort. We are currently working on extending our international collaborations to allow histology-specific models in the future. 

The cohorts used in this study were retrospectively gathered from two different medical centers. For the training cohort, patients were treated in the department of radiation oncology and the department of orthopedic surgery which led to a relatively high number of low-grade STS. The testing set was derived only from a radiation oncology department, leading to an overall lower number of low-grade STS. Overly aggressive STS with a metastatic state may thus be underrepresented at first diagnosis. As a consequence, in the future, non-invasive grading models should be tested in less biased cohorts.

This work bears several limitations. Both study cohorts were collected retrospectively, constituting a reason for a potential source of bias as described above [54]. Due to the multicentric setting, the patient cohorts presented a large technical heterogeneity, including different imaging protocols and MRI scanner types. Despite this heterogeneity, successful reproduction of CNN models was possible, showing effective generalizability. In our work, we compared the performance of two imaging sequences. To ensure a maximum amount of information, we used all available imaging studies per sequence leading to slightly different sizes of the training set. Relative underrepresentation in the training set of T2FS may have impaired a better classification performance. Moreover, due to sequence availability from both centers, our analysis was restricted to only fat-saturated MRI sequences. As fat-dependent signals constitute important semantic features, other sequences such as T2-weighted could provide complementary information. 

## 5. Conclusions

In conclusion, we demonstrated that both MRI-based DL models were able to classify tumor grading in soft-tissue sarcoma patients. Attention maps can provide insight into semantic imaging features relevant for model classification and can function as a valuable tool for patient-specific quality assurance. Further investigation is warranted to establish imaging-based biomarkers for non-invasive STS characterization. 

## Figures and Tables

**Figure 1 cancers-13-02866-f001:**
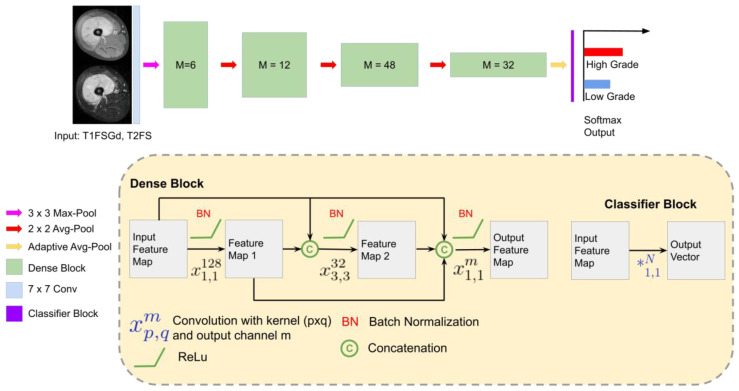
Deep learning strategy: DenseNet 161 architecture for tumor grading in MRI [44]. The network receives the 2D transversal slice from the VOI and outputs the probability of the image for the two classes. In the lower part of the figure, each component of the DenseNet is described. Abbreviations: Avg: Average, T1FSGd: contrast-enhanced and fat-saturated T1-weighted sequence, T2-weighted fat-saturated (T2FS) sequence.

**Figure 2 cancers-13-02866-f002:**
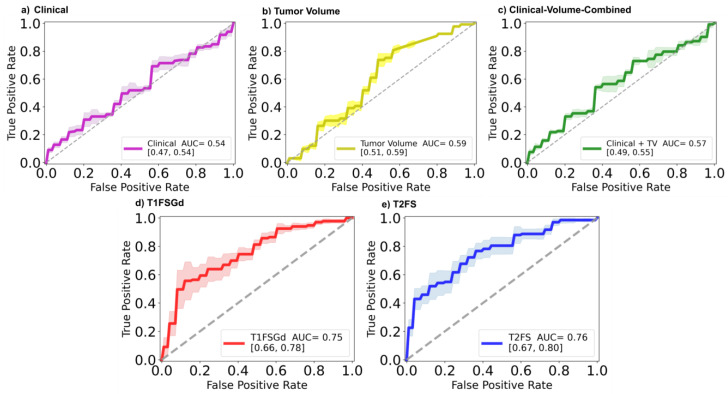
Predictive performance of MRI-based DL models. Receiver operator characteristic curves (ROC) and the respective area under the curve (AUC) values depicting the performance of the prediction models (**a**) *Clinical*, (**b**) *Tumor-Volume*, (**c**) *Clinical-Volume-Combined*, (**d**) *DL-T1FSGd*, and (**e**) *DL-T2FS*.

**Figure 3 cancers-13-02866-f003:**
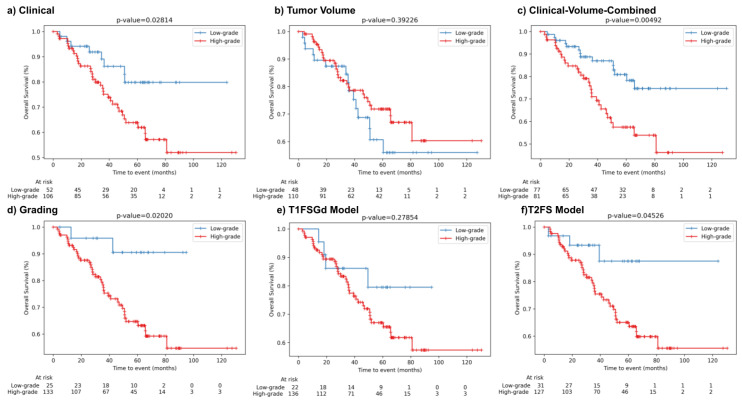
Patient risk stratification in the independent test set. Kaplan Meier survival curves for patients’ overall survival displaying risk stratification of the developed models on the test cohort. (**a**) *Clinical*, (**b**) *Tumor-Volume*, (**c**) *Clinical-Volume-Combined*, (**d**) *Grading* (low-grade vs. high-grade), (**e**) *DL-T1FSGd*, and (**f**) *DL-T2FS*. Depicted *p*-values describe the results of the log-rank test.

**Figure 4 cancers-13-02866-f004:**
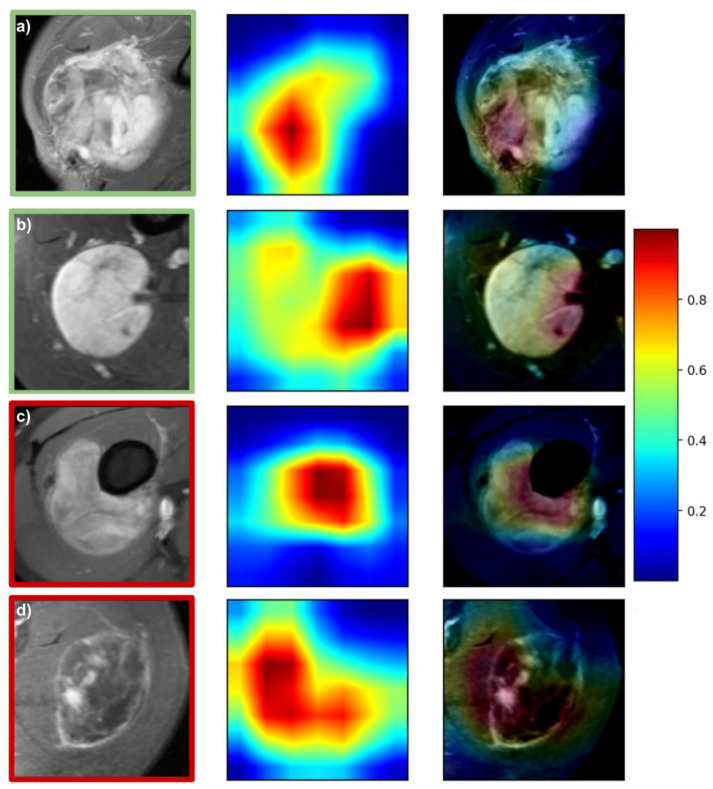
Attention maps of the *DL-T1FSGd* model. Green and red squares around images denote correct and false predictions, respectively: (**a**) correct prediction with 87% probability: high-grade (G2) synovial sarcoma—focus on tumor texture and tumor-tissue border with low confinement; (**b**) correct prediction with 95% probability: low-grade (G1) myxoid liposarcoma—focus on tumor-vessel interface with good confinement. (**c**) False prediction with 87% probability: low-grade (G1) myxoid liposarcoma—focus on tumor-bone interface. (**d**) False prediction with 98% probability: high-grade (G3) pleomorphic sarcoma—focus on central tumor parts and tumor-tissue border, low in-plane resolution.

**Figure 5 cancers-13-02866-f005:**
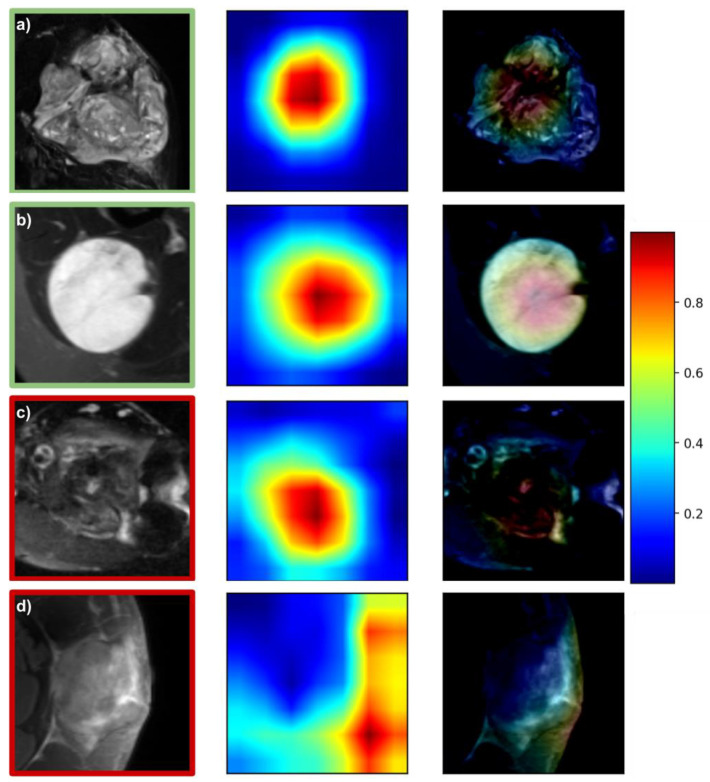
Attention maps of the *DL-T2FS* model. Green and red squares around images denote correct and false predictions, respectively: (**a**) correct prediction with 99% probability: high-grade (G3) spindle cell sarcoma—focus on tumor texture; (**b**) correct prediction with 97% probability: low-grade (G1) myxoid liposarcoma—focus on tumor texture; (**c**) false prediction with 52% probability: low-grade (G1) myofibrosarcoma—focus on tumor texture; (**d**) false prediction with 97% probability: high-grade (G3) pleomorphic sarcoma—trunk location, focus on air.

**Table 1 cancers-13-02866-t001:** Patient demographics and outcome.

Institution		TUM	UW	*p*-Value ^1^
Total Patients		148 p	158 p	
Location				1
	Extremity or trunk	141/148 p (95.2%)	154/158 p (97.4%)	
	Abdomen/retroperitoneal	5/148 p (3.3%)	2/158 p (1.3%)	
	Thorax	1/148 p (0.6%)	0/158 p (0%)	
	Head and neck	1/148 p (0.6%)	2/158 p (1.3%)	
Age		57.29 ± 17.48	53.91 ± 15.40	0.04 *
Gender				
	Female	69/148 p (46.6%)	95/158 p (60.2%)	0.2
	Male	79/148 p (53.4%)	63/158 p (39.8%)	
T-Stage				
	1	25/148 p (16.8%)	28/158 (17.7%)	0.88
	2	123/148 p (83.2%)	130/158 p (83.3%)	
	a	13/148 p (8.7%)	6/158 p (3.7%)	0.09
	b	135/148 p (91.3%)	152/158 p (96.3%)	
M-Stage				
	0	140/148 p (94.6%)	153/158 p (96.8%)	0.40
	1	8/148 p (5.4%)	5/158 p (3.2%)	
N-Stage				
	0	145/148 p (98%)	158/158 p (100%)	0.11
	1	3/148 p (2%)	0/158 p (0%)	
Grading ^2^				0.16
	1	52/148 p (35.1%)	25/158 p (15.8%)	
	2	36/148 p (24.4%)	53/158 p (33.6%)	
	3	60/148 p (40.5%)	80/158 p (50.6%)	
Tumor volume		294.52 ± 442.07	320.0 ± 487.04	0.42
AJCC-Stage ^3^				0.47
	IA	10/148 p (6.7%)	5/158 p (3.1%)	
	IB	42/148 p (28.3%)	20/158 p (12.6%)	
	IIA	11/148 p (7.4%)	23/158 p (14.5%)	
	IIB	5/148 p (3.3%)	37/158 p (23.4%)	
	III	72/148 p (48.6%)	68/158 p (43.0%)	
	IV	8/148 p (5.4%)	5/158 p (3.16%)	
Median OS		37.37 mo	45.8 mo	0.25
Available imaging				
T1FsGd		148	158	
T2FS		130	158	

Abbreviations: *: *p*-value < 0.05, AJCC: American Joint Committee on Cancer and the International Union for Cancer Control, m: median, p: patients, r: range, RT: radiation therapy. ^1^ Wilcoxon rank-sum test for continuous and ordinal variables, Fisher’s exact test for nominal variables, log-rank test for comparison of survival times. Corrected for multiple testing by Bonferroni correction (“*p*-value adjusted”). ^2^ According to the French Federation of Cancer Centers Sarcoma Group (FNCLCC). ^3^ Following AJCC staging system version 7 [50].

**Table 2 cancers-13-02866-t002:** Classification metrics for the test set. In bold, the best result among all models for each metric is marked.

	Precision	Sensitivity	Specificity	F1-Score	Accuracy
***Clinical***	0.87	0.69	0.44	0.77	0.65
***Tumor Volume***	0.89	0.74	0.52	0.81	0.70
***Clinical-Volume-Combined***	0.89	0.54	0.64	0.67	0.56
***DL-T1FsGd***	0.89	**0.91**	0.40	**0.90**	**0.83**
***DL-T2Fs***	**0.92**	0.62	**0.72**	0.74	0.64

## Data Availability

The data presented in this study are available on request from the corresponding author dependent on ethics board approval. The data are not publicly available due to data protection legislation.

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
