# Peer review of "Development and External Validation of Deep-Learning-Based Tumor Grading Models in Soft-Tissue Sarcoma Patients Using MR Imaging"

_cancers, 2021, doi:10.3390/cancers13122866_

Round 1

Reviewer 1 Report

Review:

Development and External Validation of Deep-Leaning Based Tumor grading Models in Soft-

Tissue Sarcoma Patients using MR imaging.

Navarro et al.

The authors present a creation of a deep learning model to predict grade in soft tissue sarcomas by using MRI.  The authors used two independent cohorts (TUM and UW).  Data was collected retrospectively.  All soft tissue sarcomas were included.  Exclusion criteria were use of megasprosthesis/megaprosthesis artifact on MRI, Primary bone sarcomas, Ewing Sarcoma, History of radiation.  Cohorts were dichotomized as low-grade tumors and high grade/intermediate grade tumors according to the final histology.

The two used sequences included: T1FSGd in 148 patients and T2FS in 130 patients.  This describes the training set.  In the test cohort only patients that had both sequences available were included: 158 patients in total.  After processing and comparison with clinical models, the authors conclude that the DL models based on MRI were better in predicting grade and separating survival into low and high-risk patients.  The only sequence that reached significance was DL-T2FS

The introduction is well written and well structured.  I think it is important to clarify that deep learning is a type of artificial intelligence in which the code allows for self-training of the process whereas that is not the case for all modalities of artificial intelligence.   Also the number of events of evaluation differ.  A lot of observation or events are necessary in deep learning (thousands) whereas other model of AI do not need as many observations.  The authors are clear in that they are using deep learning.  My guess is that this is because of all the observations that are being done per case. This is important to explain to the clinical/general reader.

Materials and methods are well organized.  Manual segmentation and subsequent methods are explained.  Edematous changes are not included in the segmentation.  This is a possible flaw as in some tumors such myxofibrosarcomas or dermatofibrosarcomas these edematous changes actually include tumoral cells.  It is not possible to not know a priori of these changes contain or not tumor.  I would suggest that either myxofibrosarcomas and highly infiltrative tumors are excluded or analyzed independently with segmentation including the edematous areas or are simply excluded. 

Two independent DL models were created for each type of sequence. Data was processed with DenseNet 161 which demonstrated to have the optimal architecture for tumor grading when compared to shallower and deeper pre-trained networks.  All models were developed with Pytorch with a 12  GB titalk XP.  The obtained 15 optimized models were tested in the UW cohort.  Subsequent comparison between DL models and  logistic regression of clinical characteristics as used for clinical validation. 

Results are clear and well presented.  I like that the present accuracy but take into consideration precision and less imbalance-biased metric which is not usually reported.       

The discussion is well written and highlights my concerns such as the heterogeneity in terms of tumors.  To expect to do this study with each histology is unrealistic but I think that grouping attempting to make the cohorts more homogenous is worth training.  I would suggest to try the model in tumor with high myxoid matrix or in fibrous matrix tumors.  I think the issue of segmentation not including the edemaous areas may be relevant in tumors such as myxofibrosarcoma.  They point out the issues of grade in different histologies which is a concerning area.  It is good that these issues are discussed but by discussing then they do not go away and still affect the quality of the paper.

This is an interesting concept and I like that it generalizable but more uniformity should be attempted to improve performance and clinical relevance.

I recommend publication with some modifications

Reviewer 2 Report

A DL-based approach for STS grading is novel and clinically relevant. But, there are 2 major questions that need to be addressed.

  1. The patient demographics need to be clarified.

1) The huge difference in % of low grade tumors between the 2 cohorts need to be clarified.

2) In Table 1 UW cohort, the numbers don’t match (grade 1s (n=23) vs. AJCC IA 3+ IB 14 (n=17))

3) Table S1 is not available for review.

  1. The baseline models are not adequately built. 

1) I am not sure how the authors utilized 4 parameters (T-stage, N-stage, M-stage, Age) to examine “clinical” baseline model.

2) By intuition, “clinical model” should perform better than “tumor volume model”, as N-stage and M-stage is obviously associated with aggressive biology, thus higher grades (Figs 2 and 3).

Round 2

Reviewer 1 Report

The authors have addressed the points raised during the revision process.  The authors have provided sound responses.  I recommend the manuscript to be published in its current form.